# Gut Microbes and Neuropathology: Is There a Causal Nexus?

**DOI:** 10.3390/pathogens11070796

**Published:** 2022-07-14

**Authors:** Katherine Dinan, Timothy G. Dinan

**Affiliations:** 1School of Medicine, Trinity College, DO2PN40 Dublin, Ireland; dinank@tcd.ie; 2APC Microbiome Ireland and Department of Psychiatry, University College Cork, T12CY82 Cork, Ireland

**Keywords:** brain-gut-microbiota axis, probiotics, psychobiotics, prebiotics, faecal microbiota transplantation, neurological disorders

## Abstract

The gut microbiota is a virtual organ which produces a myriad of molecules that the brain and other organs require. Humans and microbes are in a symbiotic relationship, we feed the microbes, and in turn, they provide us with essential molecules. Bacteroidetes and Firmicutes phyla account for around 80% of the total human gut microbiota, and approximately 1000 species of bacteria have been identified in the human gut. In adults, the main factors influencing microbiota structure are diet, exercise, stress, disease and medications. In this narrative review, we explore the involvement of the gut microbiota in Parkinson’s disease, Alzheimer’s disease, multiple sclerosis and autism, as these are such high-prevalence disorders. We focus on preclinical studies that increase the understanding of disease pathophysiology. We examine the potential for targeting the gut microbiota in the development of novel therapies and the limitations of the currently published clinical studies. We conclude that while the field shows enormous promise, further large-scale studies are required if a causal link between these disorders and gut microbes is to be definitively established.

## 1. Introduction

The adult human brain weighs around 1.5 kg, approximately the same as the gut microbiota. The number of bacteria in the microbiota is similar to the number of cells in the human body [1]. However, there is greater DNA content with a larger number of genes in the microbiota than in the human cell. Humans and microbes have co-evolved in a symbiotic relationship, and while we feed the microbes, in turn, they produce molecules that our brains and other organs require for survival. If all the microbial DNA required by us was directly incorporated in our cells, there would be inadequate space for such material. Thus, the symbiotic relationship is essential for our physiological functioning. Any significant derangement of the gut microbiota architecture can impact brain function and result in neuropathology [2]. 

## 2. Microbiota Structure

Advances in DNA sequencing and bioinformatics have enabled an increased understanding of gut microbiota architecture and function [3]. Bacteria are by far the dominant microbe within the microbiota architecture, but other microbes include viruses, fungi and archaea [4]. Currently, there are 25 recognised bacterial phyla, which include Firmicutes, Bacteroidetes, Actinobacteria, Cyanobacteria, Fusobacteria, Proteobacteria and Verrucomic. It is estimated that Bacteroidetes and Firmicutes phyla account for around 80% of the total human gut microbiota. The genera within the Firmicutes phylum include *Clostridium, Lactobacillus*, *Bacillus*, *Clostridium*, *Enterococcus* and *Ruminicoccus*. The Bacteroidetes phylum predominantly consists of the *Bacteroides* and *Prevotella* genera. The Actinobacterium phylum is dominated by the *Bifidobacterium* genus [5].

Approximately 1000 species of bacteria have been identified in the human gut. There remains a lack of consensus as to the structure of a normal gut microbiota and hence as to a definition of dysbiosis [6]. This lack of consensus is due to the significant inter-individual differences seen in the microbiota structure. However, it is generally accepted that greater microbial diversity in adulthood together with high levels of resistance and resilience are associated with a healthy state. Furthermore, some genera are classified as beneficial symbionts, while others have been classified as potential pathogens. An altered ratio of symbionts to pathogens may increase disease vulnerability. Bifidobacteria and lactobacilli are generally viewed as beneficial bacteria and are frequently sold as probiotic supplements (live biotherapeutics). Strains within the *Clostridium* genus or lipopolysaccharide (LPS)-forming taxa such as Enterobacteriaceae have been associated with disease states [7]. Though somewhat controversial, the Firmicutes/Bacteroidetes ratio, exploring the relationship between the two dominant phyla, has been connected with various pathologies [8]. It has been argued that exploring the functional state of the gut microbiota is of greater value in studying disease than describing the overall architecture.

In adulthood, the core gut microbiota is relatively stable. The main factors influencing structure are diet, exercise, stress, disease and medications [9]. It is generally accepted that a Mediterranean diet rich in fruit, vegetables, nuts, fermented foods and fish yields an optimal microbiota [10]. Up to 80% of medications prescribed by clinicians impact the gut microbiota, and many in a negative manner [11]. Broad-spectrum antibiotics have a dramatic impact, but after such treatment, the microbiota does gradually revert to a pre-antibiotic state. This may take up to twelve weeks. 

The gut microbiota decreases in diversity with ageing [12]. Healthy elderly people tend to maintain microbial diversity, unlike those who age in an unhealthy manner. These significant shifts in microbiota composition may be at the heart of many elderly disease processes, including those involving the brain.

## 3. Microbe to Brain Communication

All the important neurotransmitters can be produced by microbes [13]. These neurotransmitters include GABA, norepinephrine, dopamine, serotonin and histamine. GABA, for instance, is produced to varying degrees by all lactobacilli [14]. However, even when produced by gut bacteria, it is highly unlikely that these neurotransmitters travel to and influence the brain. However, they may affect the enteric nervous system and other brain communication networks such as the vagus nerve. In the case of serotonin, it has been shown that bifidobacteria increase plasma tryptophan levels [15]. The latter is the building block of serotonin, and the human brain has limited storage capacity. It seems likely that diet and bifidobacterial synthesis play an important role in maintaining normal central serotonergic function. If diet was the sole source of tryptophan, it is likely that several days of starvation would have a profound effect on mood. In such circumstances, microbial production provides an important alternative to dietary tryptophan.

The routes of communication between gut microbes and the brain are not fully elucidated but do include neural, endocrine, immune and metabolic pathways (Figure 1) [16]. The vagus nerve, a long meandering nerve, is a major route of bidirectional information signalling between the gut and the brain [17]. Several important chemicals are produced in the gut and travel to the brain via the bloodstream. Many of these are manufactured by or have their production regulated by gut microbes. These include short-chain fatty acids, tryptophan, leptin and ghrelin. The short-chain fatty acids include butyrate, propionate and acetate. Butyrate is a known inhibitor of histone deacetylases (HDACs), and by doing so, acts as an epigenetic regulator [18]. It also has the ability to influence the G-protein coupled receptors, namely free fatty acid receptors [19], though these are sparsely distributed in the mammalian brain. Cytokines, key immune regulatory molecules produced at the level of the gut, can travel via the bloodstream and influence brain function, especially in brain regions where there is a deficient blood-brain barrier, such as the hypothalamus.

Over the past decade, several groups have examined the gut microbiota in a variety of neurological disorders, viewing such conditions as disorders of the brain-gut-microbiota axis, and there are several published reviews [21,22,23,24]. However, a limitation in most of these clinical studies is a failure to control for relevant variables such as diet, levels of exercise and medication usage. The impact of diet on the gut microbiota can be a significant confounder if not adequately controlled. This was clearly illustrated in the recent autism study by Yap et al. [25]. Here, we will examine the findings reported in some of the major neurological conditions. Most of the published reviews come from a preclinical perspective. In contrast, the current review begins from the angle of the clinician and translates back to the preclinical findings. We focus on Parkinson’s disease, Alzheimer’s disease, multiple sclerosis and autism because of their high prevalence in western society. Such a focus is not meant to diminish the studies in other therapeutic areas such as motor neurone disease and Huntington’s disease. 

The selected studies were based on a PubMed search of the following terms: gut microbiota, brain-gut-microbiota axis and each of the individual diseases. The analysis is based on papers published until June 2022. Preclinical studies were chosen on the basis that they provided potential leads to clinical pathophysiology. 

## 4. Parkinson’s Disease

Parkinson’s disease (PD), the ‘shaking palsy’, is named after James Parkinson, the East London surgeon who provided the first detailed description of the disorder in 1817 in his paper “An Essay on the Shaking Palsy”. He was probably not the first to describe the disease, which may originally have been recognised by the Roman physician Galen. The term Parkinson’s disease was coined by the neurologist Jean-Martin Charcot. The condition is characterised by slow movements (bradykinesia), resting tremor, rigidity and postural instability, and can profoundly impact quality of life. Its underlying pathology is degeneration of dopamine neurones in the zona compact region of the substantia nigra. Current therapies largely focus on enhancing dopamine transmission in the nigrostriatal pathways. However, while initially helpful in managing symptoms, such treatments usually lose efficacy over time. Alternative therapies are clearly required, and the gut microbiota is viewed by many as a suitable therapeutic target.

It is argued that PD occurs because of toxins produced within the gut by microbes, or alternatively as a failure of the microbiota to manufacture neuronal dopamine-specific nutrients. 

The protein alpha-synuclein has been implicated in the pathogenesis of PD. It is speculated that aggregation of alpha-synuclein is central to the pathophysiology of the disorder and that the dopaminergic neurones in the substantia nigra are majorly sensitive and as a result degenerate with such accumulation [26]. Evidence is accumulating to support the view that the alpha-synuclein originates in the gut and spreads to the brain like a prion, probably spreading along the vagus nerve. The molecule can form aggregates, regarded as disease biomarkers, and may be the vehicle for the spread of disease from the peripheral to the brain. Furthermore, alpha-synuclein is viewed as being involved in the autonomic dysfunction seen in many patients with PD. It may also play a role in the gastrointestinal symptoms frequently seen in the condition. 

Vagotomy was a common surgical procedure for the treatment of peptic ulcers prior to the introduction of H_2_ antagonists and proton pump inhibitors. As a treatment, it was used for decades, and for many patients, it did improve symptoms, although not entirely without side effects. Most of the patients who underwent such surgery and who are still alive are now elderly. A Swedish register study [27] investigated the risk of PD in patients who underwent vagotomy and hypothesised that truncal vagotomy is associated with a protective effect. They found that the risk of PD was indeed decreased in patients who underwent full truncal vagotomy, while there was no risk reduction in those who underwent a super-selective vagotomy. Risk of PD is also decreased after truncal vagotomy when compared to the general population. These epidemiological findings are important and raise several issues requiring further exploration. Overall, the data support the hypothesis that PD commences in the gut and not initially in the brain. This provides further evidence for the involvement of the vagus nerve in the development of the disorder. 

To characterise the expression of alpha-synuclein in the innervation of the gut, both the postganglionic neurones and the preganglionic projections, by which the disease is postulated to retrogradely invade the brain, were examined [28]. It was found that some vagal preganglionic efferents expressing alpha-synuclein formed varicose terminal rings surrounding myenteric plexus neurones that were also positive for the protein. This fundamental finding provides a plausible alpha-synuclein-expressing pathway, enabling the retrograde transport of PD pathogens from the enteric nervous system to the brain. 

The gut microbiome in PD has been investigated in several research settings. For a comprehensive review, see [29]. Sheperjans and colleagues compared the faecal microbiomes of 72 PD patients and 72 control subjects using 16S ribosomal RNA sequencing [30]. They explored the associations between clinical parameters and microbiota bioinformatics. They found that the abundance of Prevotellaceae in faeces of PD patients was reduced by over 70% as compared with controls. The relative abundance of Prevotellaceae of 6.5% or less had 86% sensitivity and 39% specificity for PD diagnosis. A discriminant analysis based on the abundance of four bacterial families and constipation severity identified PD patients with 67% sensitivity and 90% specificity. Of note is the fact that the relative abundance of Enterobacteriaceae was positively correlated with the severity of postural instability and gait difficulties. These findings indicate that the intestinal microbiome is altered in PD and is related to the motor phenotype. However, the possibility that these microbiota changes are entirely epiphenomenal cannot be entirely ruled out. The findings nonetheless raise the exciting prospect that alterations in the gut microbiota are a potential diagnostic marker.

Mazmanian’s group have added considerably to the literature. They used the alpha-synuclein overexpressing mouse to investigate the genesis of PD [31]. These mice are genetically engineered to overexpress alpha-synuclein, and they go on to develop core features of PD. This is currently the most widely used animal model of the disorder. Surprisingly, when these mice are raised germ-free, their propensity to develop motor anomalies is significantly reduced. If such animals are given a combination of short-chain fatty acids, they show microglial activation in the brain and aggregation of alpha-synuclein with the onset of motor features. The changes are prevented by treatment with the antibiotic minocycline, which acts on both Gram-positive and Gram-negative bacteria. When germ-free, alpha-synuclein overexpressing mice are given a humanised microbiota from a patient with PD, the resultant pathology is greater than that seen following transplant with the microbiota from a healthy subject. The findings suggest multiple potential lines of treatment, which include short-chain fatty acid antagonists, antibiotics and microbiota transplantation.

In a recently published study, Nishiwaki and colleagues investigated the manner in which short-chain fatty acid-producing and mucin-degrading intestinal bacteria predict the progression of early PD [32]. They report that decreases in short-chain fatty acid-producing genera, Fusicatenibacter, Faecalibacterium and Blautia, as well as an increase of the mucin-degrading genus Akkermansia, predicted accelerated disease progression. This requires replication, but if replicated may be an important finding, helping the development of new innovative therapies.

Xue et al. examined the impact of faecal microbiota transplant therapy in patients with PD [33]. Fifteen PD patients were included: ten received FMT via colonoscopy and five received FMT via a nasal-jejunal tube. Overall, an improvement in both motor and psychological symptoms was reported. The colonic FMT group showed significant improvement and longer maintenance of efficacy compared with nasointestinal FMT. None of the patients were satisfied with nasointestinal FMT for more than three months. Overall, adverse events were mild. This was a preliminary open-label study which certainly provides support for larger-scale, double-blind interventions.

Tamtaji et al. [34] conducted a randomised, double-blind, placebo-controlled trial, in 60 people with PD. Subjects were randomly divided into two groups. Group 1 received a probiotic at 8 × 10^9^ CFU/day, and Group 2 received a placebo (n = 30 in each group) for 12 weeks. Consumption of the probiotic decreased neurological symptoms relative to those seen in the placebo group. The probiotic supplementation also reduced high-sensitivity C-reactive protein, insulin and malondialdehyde, and increased glutathione levels. The authors conclude that the therapy has potential benefits for patients with PD.

In summary, the gut microbiota shows promise as a potential target for intervention in patients with PD. See Table 1.

## 5. Alzheimer’s Disease (AD)

AD is the most common cause of dementia and occurs in a senile and pre-senile form. The latter begins before the age of sixty-five, and the former occurs after this age. It was originally described in 1906 by the pathologist Alois Alzheimer and we now know that the neuropathology includes amyloid plaques and neurofibrillary tangles [44]. The World Health Organization (WHO) estimates that over 50 million people worldwide suffer from dementia, with AD accounting for 60–70% of all cases. In the USA alone, it costs the healthcare system over USD250 billion annually.

Germ-free mice who lack a gut microbiota have a marked absence of amyloid plaque and neuroinflammation. Transgenic mouse studies of AD have shown an altered microbiome [45,46]. AD histological and cognitive features in the APP/PS1 mouse model of AD correlated with gut microbiome alterations. Specifically, Helicobacteraceae and Desulfovibrionaceae at the family level and Odoribacter and Helicobacter at the genus level are increased relative to controls [47]. 

When APP/PS1 transgenic mice are treated with an antibiotic cocktail, a reduction in microglial and astrocyte accumulation surrounding amyloid plaques is observed. Furthermore, modulation of intestinal microbiota by the probiotic VSL#3 resets brain gene expression and ameliorates the age-related deficit [48]. With such animal findings, there is now an increasing focus on dietary interventions targeting the microbiome for the management of AD. In an animal model of AD, administration of *Bifidobacterium bifidum* and *Lactobacillus plantarum* for eight weeks with physical training had a positive effect. A decrease in amyloid-β protein and an improvement in spatial learning via an acetylcholine-mediated mechanism were found.

There are no prospective studies of the gut microbiota in patients with AD published so far. Cattaneo et al. [49] conducted a cross-sectional study which identified Escherichia/shigella bacterial taxa as increased in faecal samples from AD patients relative to control subjects. The data suggest that these taxa are associated with inflammation. Of note is the fact that microbiota changes correlate with pro-inflammatory blood cytokines levels. The authors suggest a causal link between dysregulation of the microbiota and systemic inflammation and speculate that such microbiota changes initiate or exacerbate the neurodegeneration in AD. See Table 2.

In a recent study, Wang et al. compared faecal microorganisms of patients with AD and healthy cohabiting caregivers [50]. They found that the abundance of several bacteria taxa in AD was altered at the genus level. The most notable changes were Anaerostipes, Mitsuokella, Prevotella, Bosea, Fusobacterium, Anaerotruncus, Clostridium and Coprobacillus. Surprisingly, Akkermansia, an available probiotic, was increased in the AD patients compared with controls. Additionally, the levels of Bifidobacteria were found elevated. The authors conclude that “patients with mild AD have unique gut microbial characteristics”. Such changes may present a novel target for therapeutic intervention.

Asaoka et al. [51] conducted a randomised clinical trial on 130 patients aged from 65 to 88 years old with mild cognitive impairment. They received either a daily probiotic (*B. breve* MCC1274, 2 × 1010 CFU) or a placebo for 24 weeks. The probiotic intervention produced marked changes relative to the placebo. The data suggest that the intervention suppressed brain atrophy progression and ameliorated cognitive deterioration. Interestingly, the composition of the microbiota was not changed by the intervention. 

**Table 2 pathogens-11-00796-t002:** Microbiota changes in Alzheimer’s disease.

	Reference
*Lachnoclostridium, Bacteriodes*	[52]
*Enterococcaceae. ↑*	[52]
*Akkermansia, Blautia, Dorea, Eggerthella, Streptococcus, Bifidobacterium, Lactobacillus ↑*	[53]
*Alistipes, Bacteroides, Butyricimonas, Haemophilus, Parabacteroide ↓*	[53]
*Enterobacteriaceae, Veillonellaceae ↑*	[54]
*Clostridiaceae, Lachnospiraceae, RuminococcaceaeGenus: Blautia, Ruminococcus ↓*	[54]
*Bacteroidaceae, Rikenellaceae, GemellaceaeGenus: Blautia, Bacteroides, Alistipes, Bilophila, Gemella, Phascolarctobacterium ↑*	[55]
*Ruminococcaceae, Bifidobacteriaceae, Clostridiaceae, Peptostreptococcaceae, Mogibacteriaceae, Turicibacteraceae Genus: Bifidobacterium, Dialister, Clostridium, Turicibacter, Adlercreutzi ↓*	[55]
*Escherichia, Shigella ↑*	[49]

## 6. Multiple Sclerosis (MS)

MS is an autoimmune disorder associated with the demyelination of neurones in the brain and spinal cord [56]. The most common presentation shows a relapsing/remitting pattern, but some patients present with a more severe form of primary progressive disorder. The symptoms are multitudinous but usually involve movement, sensation or balance. Despite therapeutic advances, the condition can have a profound negative impact on quality of life. 

Germ-free mice show increases in myelination patterns in the prefrontal cortex, suggesting that the gut microbiota plays a role in controlling myelination [57]. Furthermore, germ-free mice are especially resistant to developing experimental autoimmune encephalomyelitis (EAE), which is a well-established animal model of MS. Other studies indicate that the presence of certain Gram-positive segmented filamentous bacteria, known to activate Th17 cells, significantly alter EAE severity [58]. 

Navarro-López [59] analysed the gut microbiota of 15 patients with relapsing/remitting MS and compared samples with those from diet-matched healthy controls. They found that the gut microbiota of the MS group differed significantly from healthy controls in the levels of the Lachnospiraceae, Ezakiella, Ruminococcaceae, Hungatella, Roseburia, Clostridium, Shuttleworthia, Poephyromonas and Bilophila genera. The data provide support for the view that MS is associated with a dysbiosis and that Ezakiella and Bilophila genera in particular may be a risk factor in patients with relapsing/remitting MS. See Table 3.

FMT has been assessed in a preliminary study of patients with MS [60]. Nine patients with MS were recruited and underwent FMTs monthly for up to six months. The primary outcome measure was alteration in blood cytokines. The secondary outcomes were gut microbiota composition together with intestinal permeability. Microbiota enrichment was observed, and in two patients who had increased intestinal permeability at baseline, the FMT decreased the permeability. The study was preliminary and obviously limited by the small sample size. The overall conclusion is that FMT is safe in patients with MS. 

Mirashrafi et al. [61] conducted a meta-analysis of probiotic trials where probiotics were used as adjunctive therapy in patients with relapsing/remitting MS. A total of 106 patients were randomly assigned to a probiotic treatment, and 107 were randomly assigned to a placebo. The probiotics used included *Lactobacillus acidophilus*, *Lactobacillus casei*, *Bifidobacterium bifidum*, and *Lactobacillus fermentum*, *Bifidobacterium infantis*, *Bifidobacterium lactis* and *Lactobacillus reuteri*. A significant improvement with probiotic supplementation was found in the Expanded Disability Status Scale as well as on psychological measures. This meta-analysis is of interest but limited by the large variety of probiotics used and the different durations of therapy. A large-scale study with the most promising probiotic needs to be conducted.

**Table 3 pathogens-11-00796-t003:** Microbiota changes in multiple sclerosis.

	References
*Dorea, Pedobacter, Flavobacterium,*	
*Prevotella, Parabacteroides, Collinsella ↓*	[62]
*Methanobrevibacter, Akkermansia ↑*	[63]
*Butyricimonas, Prevotella ↓*	
*Prevotella ↓*	[64]
*Streptococcus ↑*	[65]
*Prevotella, Faecalibacterium ↓*	
*Lawsonella ↑*	[66]
*Faecalibacterium Prausnitzii,*	
*Bacteroides fragiils, Eubacterium rectale,*	
*Bbutyrivibrio, Clostridium, Coprococcus, Roseburia ↓*	
*Faecalibacterium, Ruminococcus ↑*	[21,67]
*Blautia, Anaerostipes, Bifidobacterium, Prevotella*	

## 7. Autism

Autism is a disorder with an early age of onset, characterised by poorly developed social interactions, difficulty with receptive and expressive communication and the presence of repetitive behaviours [68]. Epidemiological evidence suggests that the prevalence of the disorder is on the rise, though this may represent changes in diagnostic patterns. Gastrointestinal disturbance is frequently co-morbid. Constipation is the most common symptom, followed by abdominal pain, diarrhoea and occasionally vomiting. Overall, the condition is more common in males than females, and in general autistic children grow into autistic adults, though early behavioural interventions can ameliorate symptoms. 

In a recent paper, Lu et al. [69] argue that rebalancing the gut microbiota is the critical step in treating autism and the broader autism spectrum disorder (ASD).

Germ-free studies provide support for the link between gut microbiota and autism. Such animals show behavioural changes similar to in autism [70]. When allowed to interact with another mouse or an inanimate object, germ-free mice are as likely to interact with the latter. Such behaviour is abnormal for socially interactive animals. If germ-free animals undergo conventional colonisation at an early stage, the emergence of behavioural changes can be prevented. In fact, the administration of single bacterial strains such as *Bacteroides Fragilis* or *Lactobacillus reuteri* can reverse many of the behavioural changes in these animals [71].

Numerous types of dietary interventions have been employed in treating autism, including the Mediterranean diet, gluten-free diet and ketogenic diet. For a review, see [72]. Here, we will focus our attention on directly impacting microbial interventions such as probiotics, which are sometimes referred to as psychobiotics when associated with the management of syndromes such as this with a behavioural component. A psychobiotic is defined as bacteria that, when ingested in adequate amounts, has a positive mental health benefit [73]. An intervention study found that *Lactobacillus plantarum* WCSF1 increased the number of *Lactobacillus* and *Enterococcus* bacteria in autism, and significantly decreased the level of *Clostridium cluster* XIVa, viewed as a harmful bacterium. More importantly, from the perspective of the patient, after the probiotic, the scores of destructive behaviour, anxiety, self-focused behaviour and poor communication were improved. A different strain of *Lactobacillus plantarum* (PS128) was shown to reduce the scores on the social responsiveness scale (SRS) and the clinical global impression (CGI) scale [74]. *Lactobacillus acidophilus* is also found to improve the concentration of ASD children. See Table 4.

Some groups have studied synbiotics, the combination of probiotic and prebiotic fibre. The latter promotes the growth of good bacteria. One month of treatment with the polybiotic combination *Bifidobacterium infantis* Bi-26, *Lactobacillus rhamnosus* HN001, *Bifidobacterium lactis* BL-04 and *Lactobacillus paracasei* LPC-37, together with fructooligosaccharide, brought about improvements in speech/language and overall social interaction [75]. This observation is supported by a recent systematic review which found that supplementing with multiple probiotic species or the addition of a prebiotic fibre produced a better result than the use of a single strain [76]. 

Given the potential of using the gut microbiota as a target [77], it is not surprising that FMT has been used as a potential therapy. FMT was found to improve behavioural problems and gastrointestinal symptoms in younger patients but not in those over twenty-one years [69]. In a further FMT study, Zhao et al. [53] conducted a randomised controlled study on 48 patients with ASD. The study was not double-blind. The FMT group had two FMT treatments, and the control group received a behavioural intervention. After the first FMT treatment, the Childhood Autism Rating Scale (CARS) scores in the FMT group decreased by 10.8%, compared with 0.8% in the control group. After the second FMT, further improvement was observed. However, FMT was associated with some transient side effects, such as fever.

In a further study, the vancomycin antibiotic was first administered, followed by a bowel preparation and then FMT [78]. Eighteen ASD children aged 7–16 years old were included and, at baseline, were compared with twenty healthy age-matched controls. FMT not only improved their behavioural symptoms but also increased bacterial diversity with a decrease in gastrointestinal symptoms. There were increases in the abundance of Bifidobacterium, Prevotella and Desulfovibrio. The behavioural improvements were sustained for up to two years. Although the sample size is relatively small, the findings show enormous promise for FMT.

**Table 4 pathogens-11-00796-t004:** Microbiota changes in autism.

	Reference
*Clostridium and Ruminococcus* spp. *↑*	[79]
*Bacteroidetes, Proteobacteria, Alkaliflexus, Desulfovibrio, Acetanaerobacterium, Bacteroides, Parabacteroides, Desulfovibrio* spp. *↑* *Actinobacteira, Turicibacter Clostridium, Firmicutes, Weissella, Helcococcus, Alkaliphilus, Anaerofilum, Pseudoramibacter, Ruminococcus, Streptococcus, Anaerovorax, Dialister, Lactococcus ↓*	[80]
*Firmicutes↓, Proteobacteria↓, Verrucomicrobia ↓* *Bacteroidetes/Firmicutes↑, Dialister↓* *Prevotella ↑ Bacteroides, Megamonas, Escherichia/Shigella, Lachnospiracea_incertae_sedis ↑* *Clostridium XlVa, Eisenbergiella, Clostridium IV, Flavonifractor, Haemophilus, Akkermansia ↓*	[81]
*Lachnospiraceae, Clostridiales, Dorea, Erysipelotrichaceae, Collinsella,* *Lachnoclostridium ↑* *Bacteroides, Faecalibacterium, Parasutterella, Paraprevotella ↓*	[82]
*Firmicuteses, Megamonas, Proteobacteria, Actinobacteria, Dialister, Escherichia-Shigella, Bifidobacterium ↑ Bacteroidetes ↓*	[83]
*Clostridium, Dialister, Coprobacillus ↑ Faecalibacterium ↓*	[84]

## 8. Discussion

The field of the microbiota-gut-brain axis and neurological disease is novel and in most respects, an area in the early stages of development. See Figure 2. For this reason and because of the complexity of conducting well-controlled clinical studies, the literature is skewed in favour of preclinical studies. Consequently, there are now more published speculative reviews than papers focusing directly on carefully phenotyped patient populations. Here, we have pinpointed the clinical studies required to bring the field forward. 

A greater understanding of the role of gut microbes in disease pathogenesis raises the possibility of nutritional interventions. Mohan et al., for instance, explore the potential role of dietary gluten in neurodegeneration [85]. Cortical excitability to transcranial magnetic stimulation, after a lengthy gluten-free diet, modulated the electrocortical changes in celiac disease [86]. However, the use of gluten-free diets in managing cognitive impairment remains very controversial. 

There are, however, a multitude of other nutritional interventions that could be undertaken for the aforementioned diseases. Conducting such studies on a large scale will be expensive and necessitate multiple research centres. For meaningful findings, the research needs to be powered in a similar manner to that for pharmaceutical interventions, and the studies should be double-blind, random-allocation, with a parallel-group design. Adequate powering is essential, but such large-scale non-pharmaceutical industry funding will be challenging to acquire. 

## 9. Conclusions

Enormous progress has been made, over the past decade or two, in examining the impact of the gut microbiota on brain function. As with any emerging field, there are far more data available from preclinical than clinical studies, and the latter are largely cross-sectional in nature. What is now required are prospective, longitudinal studies exploring the microbiota in well-phenotyped patients. Whether the microbiota changes are a cause or a consequence of disease has yet to be definitively shown. Furthermore, we require large-scale intervention studies with a parallel-group, double-blind, placebo-controlled design to examine the impact of microbiota-targeted interventions, including FMT. 

## Figures and Tables

**Figure 1 pathogens-11-00796-f001:**
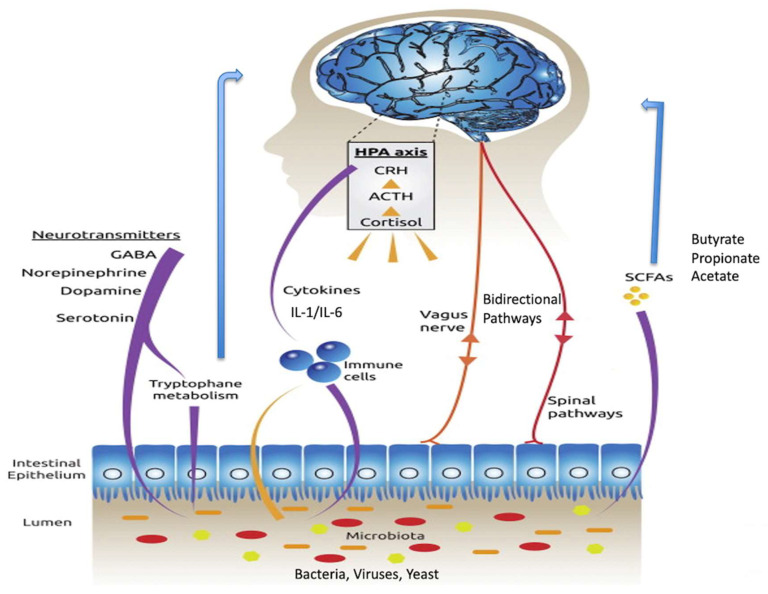
Connections within the brain-gut-microbiota axis, including the vagus nerve, and SCFAs such as butyrate, cytokines and tryptophan. Modified from Ref. [20] 2015 Springer Nature: More than a gut feeling: the microbiota regulates neurodevelopment and behavior. Neuropsychopharmacology 2015; 40: 241–242. Abbreviations: HPA, hypothalamic–pituitary–adrenal; CRH, corticotrophin-releasing hormone; ACTH, adrenocorticotropic hormone; GABA, gamma aminobutyric acid; SCFAs, short-chain fatty acids.

**Figure 2 pathogens-11-00796-f002:**
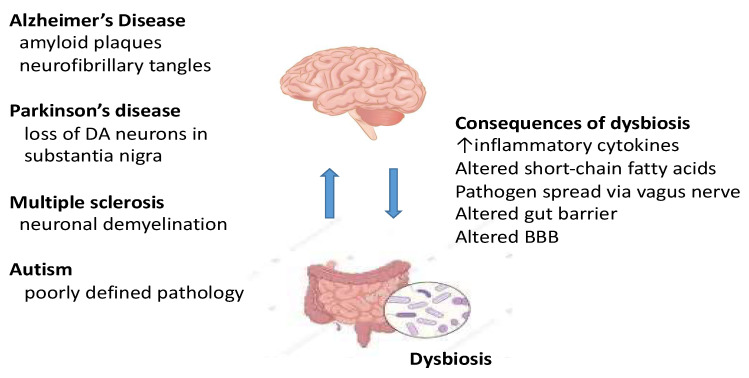
Bidirectional communication between the gut and the brain and how a gut dysbiosis can result in several central pathologies causing neurological dysfunction. Abbreviations: DA, dopamine; BBB, blood–brain barrier.

**Table 1 pathogens-11-00796-t001:** Microbiota changes in Parkinson’s disease.

	Reference
*Christensenella, Catabacter, Lactobacillus, Oscillospira, Bifidobacterium, Christensenella minuta, Catabacter hongkongensis, Lactobacillus mucosae, Ruminococcus bromii and Papillibacter cinnamivorans ↑*	[35,36]
*Dorea, Bacteroides, Prevotella, Faecalibacterium, Bacteroides massiliensis, Stoquefichus massiliensis, Bacteroides coprocola, Blautia glucerasea, Dorea longicatena, Bacteroides dorei, Bacteroides plebeus, Prevotella copri, Coprococcus eutactus and Ruminococcus callidus ↓*	[36]
*Clostridium coccoides, Bacteroides fragilis ↓*	[37]
*Roseburia, Prevotella and Bifidobacterium ↓*	[38]
Eubacterium ↓	[39]
*Lactobacillaceae, Barnesiellaceae and Enterococcacea ↑*	[40]
*Escherichia-Shigella, Streptococcus, Proteus and Enterococcus ↑* *Blautia, Faecalibacterium and Ruminococcus ↓*	[41]
*Bilophila and Paraprevotella ↑*	[42]
*Bifidobacteriaceae ↑ Lachnospiraceae ↓*	[36,43]

## Data Availability

Not applicable.

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
