# Peer review of "Gut Microbes and Neuropathology: Is There a Causal Nexus?"

_pathogens, 2022, doi:10.3390/pathogens11070796_

Round 1
Reviewer 1 Report
1. This manuscript addresses the very broad topic of potential causal roles for changes in gut microbes in neuropathology. They focus specifically on Parkinson’s disease, Alzheimer’s disease, multiple sclerosis and autism. The corresponding author has published extensively on this topic, so is very well qualified to write such a review. It is generally well written. I have some suggestions below as to how the manuscript could be improved.
2. There have been many review articles published on this topic in recent years, including by the corresponding author’s group. Below are examples of highly relevant review articles published in the past couple of years alone.
Korf JM, et al. Gut dysbiosis and age-related neurological diseases in females. Neurobiol. Dis. 2022; 168:105695. PMID: 35307514.
Anis E, et al. Digesting recent findings: gut alpha-synuclein, microbiome changes in Parkinson's disease. Trends Endocrinol. Metab. 2022; 33(2):147-157. PMID: 34949514.
Pappolla MA, et al. Indoles as essential mediators in the gut-brain axis. Their role in Alzheimer's disease. Neurobiol. Dis. 2021; 156:105403. PMID: 34087380.
Gubert C, et al. Exercise, diet and stress as modulators of gut microbiota: Implications for neurodegenerative diseases. Neurobiol. Dis. 2020; 134:104621. PMID: 31628992.
Cryan JF, et al. The gut microbiome in neurological disorders. Lancet Neurol. 2020; 19(2):179-194. PMID: 31753762.
Kadowaki A, Quintana FJ. The Gut-CNS Axis in Multiple Sclerosis. Trends Neurosci. 2020; 43(8):622-634. PMID: 32650957.
Gonzalez-Santana A, Diaz Heijtz R. Bacterial Peptidoglycans from Microbiota in Neurodevelopment and Behavior. Trends Mol. Med. 2020; 26(8):729-743.
The authors should cite recently published relevant review articles, for the information of interested readers, and make it clear near the start of their manuscript why their review is novel and necessary at this point in time.
3. Clinical studies of the microbiome in these neurological disorders are potentially confounded by the fact that patients (relative to controls) may have different diets, levels of physical activity, stress, etc. I think these potential clinical confounds, and the extent to which preclinical models have addressed the question of whether microbiome changes are intrinsic to these neurological disorders rather than secondary to lifestyle changes, should be discussed in more detail. A recent publication, indicating that autism gut microbial changes may be secondary to dietary changes, is illustrative in this regard:
Yap CX, et al. Autism-related dietary preferences mediate autism-gut microbiome associations. Cell 2021; 184(24):5916-5931.e17. PMID: 34767757.
4. The two figures are useful and informative, although Figure 2 is somewhat minimalist and could be more detailed. I think at least one or more figure would be informative, summarising potential causative links between gut microbial changes and these specific neurological disorders. I think at least one table is also needed to summarise the substantial literature in this field. Perhaps one table could summarise all of the findings on the gut microbiome in preclinical models of relevant disorders, and another table could summarise clinical findings.
5. The title of the manuscript links gut microbes to neuropathology in general, however the authors focus only on Parkinson’s disease, Alzheimer’s disease, multiple sclerosis and autism. Two other major neurodegenerative disorders where changes in gut microbes have been clearly implicated are ALS/MND and Huntington’s disease, including these articles:
Brenner D, et al. The fecal microbiome of ALS patients. Neurobiol. Aging. 2018; 61:132-137. PMID: 29065369.
Blacher E, et al. Potential roles of gut microbiome and metabolites in modulating ALS in mice. Nature 2019; 572(7770):474-480. PMID: 31330533.
Fournier CN, et al. The gut microbiome and neuroinflammation in amyotrophic lateral sclerosis? Emerging clinical evidence. Neurobiol. Dis. 2020; 135:104300. PMID: 30321601.
Kong G, et al. Microbiome profiling reveals gut dysbiosis in a transgenic mouse model of Huntington's disease. Neurobiol. Dis. 2018; 135:104268. PMID: 30194046.
Wasser CI, et al. Gut dysbiosis in Huntington's disease: associations among gut microbiota, cognitive performance and clinical outcomes. Brain Commun. 2020; 2(2):fcaa110. PMID: 33005892.
Stan TL, et al. Increased intestinal permeability and gut dysbiosis in the R6/2 mouse model of Huntington's disease. Sci. Rep. 2020; 10(1):18270. PMID: 33106549.
Kong G, et al. An integrated metagenomics and metabolomics approach implicates the microbiota-gut-brain axis in the pathogenesis of Huntington's disease. Neurobiol. Dis. 2021; 148:105199. PMID: 33249136.
Reviewer 2 Report
The authors deal with a relevant and timely topic, i.e., the exploration of a possible causal link between gut microbes and neuropathology. To this end, they narratively reviewed the literature on the involvement of gut microbiota in Parkinson’s disease, Alzheimer’s disease, Multiple Sclerosis, and Autism, also examining the potential for targeting the gut microbiota in the development of novel therapies. They concluded that, while the field shows enormous promise, further large-scale studies are required if a causal link between these disorders and gut microbes would be definitively established. Overall, the review is nicely conceived, the studies included are consistent and are adequately discussed. Few comments requiring some revision.
Abstract: please provide more details on the results reviewed; please also briefly include the translational implications of these findings and their potential clinical applications in the diagnosis and management.
Introduction: please expand this section by stating a clear aim and novelty of the present review. Moreover, the rationale underlying the selection of the above-mentioned diseases should be clarified.
Methods/Results: although the narrative design of this review, a brief “Methods” section describing the search strategy, inclusion/exclusion criteria, and selection procedures should be mentioned. The same holds true for a short “Results” section showing the number of studies retrieved, selected, and eventually included. Alternatively, the authors may succintly state how the studies here reviewed have been selected.
Discussion: before “Conclusions”, please include a short “Discussion” section briefly summarizing the main findings reviewed here and their translational applications/clinical implications. Among possible comments, recently, a dysregulation of the microbiota-gut-brain axis, the presence of specific antibodies, and the epigenetic mechanisms have been implicated in the pathogenesis linking gluten and neurodegeneration (PMID: 32751379). In this context, a “hyperexcitable celiac brain", which reverts back after a long-term gluten restriction, has been reported (PMID: 28489931). Given its potential effect on the microbiota-gut-brain axis, a gluten-free diet should be introduced as early as possible, although the overall response of neurological symptoms (and cognition in particular) is still controversial (PMID: 30065211). At the end of this new section, please also include a paragraph with limitations and future applications/research agenda.
General: please check the text for spacing, as well as the reference style (especially the in-text citations).
Round 2
Reviewer 1 Report
The authors have revised the manuscript and adequately addressed my comments.
Author Response
Thanks for your comments